# High-resolution structures of malaria parasite actomyosin and actin filaments

**Juha Vahokoski**[1], **Lesley J. Calder**[2], **Andrea J. Lopez**[1], **Justin E. Molloy**[2], **Inari Kursula**[1,3]*, **Peter B. Rosenthal**[2]*

**1** Department of Biomedicine, University of Bergen, Bergen, Norway, **2** Structural Biology of Cells and Viruses Laboratory, Francis Crick Institute, London, United Kingdom, **3** Biocenter Oulu and Faculty of Biochemistry and Molecular Medicine, University of Oulu, Oulu, Finland

\* inari.kursula@uib.no (IK); peter.rosenthal@crick.ac.uk (PBR)

**Data Availability Statement:** All structural data related to this manuscript are available in the Protein Data Bank (PDB codes 6TU4 and 6TU7) and Electron Microscopy Data Bank (EMD-10587

## Abstract

Malaria is responsible for half a million deaths annually and poses a huge economic burden on the developing world. The mosquito-borne parasites (*Plasmodium* spp.) that cause the disease depend upon an unconventional actomyosin motor for both gliding motility and host cell invasion. The motor system, often referred to as the glideosome complex, remains to be understood in molecular terms and is an attractive target for new drugs that might block the infection pathway. Here, we present the high-resolution structure of the actomyosin motor complex from *Plasmodium falciparum*. The complex includes the malaria parasite actin filament (*Pf*Act1) complexed with the class XIV myosin motor (*Pf*MyoA) and its two associated light-chains. The high-resolution core structure reveals the *Pf*Act1:*Pf*MyoA interface in atomic detail, while at lower-resolution, we visualize the *Pf*MyoA light-chain binding region, including the essential light chain (*Pf*ELC) and the myosin tail interacting protein (*Pf*MTIP). Finally, we report a bare *Pf*Act1 filament structure at improved resolution.

## Author summary

We present the structure of the malaria parasite motor complex:actin 1 (*Pf*Act1) and myosin A (*Pf*MyoA) with its two light chains. We also report a high-resolution structure of filamentous *Pf*Act1 that reveals new atomic details of the ATPase site, including a solvent-filled cavity. *Pf*Act1 undergoes no conformational changes upon *Pf*MyoA binding. The *Pf*MyoA structure in the complex superimposes with a recent crystal structure of *Pf*MyoA although small, but significant, structural rearrangements occur at the actomyosin binding interface.

## Introduction

Malaria parasites belong to the phylum *Apicomplexa*, which are obligate intracellular parasites many of which infect humans and livestock, with devastating effects on human health and mortality and large economic losses worldwide [1]. Among the best-known members of the

and EMD-10590). The optical trapping data are included as a supplemental file (S1 Data).

**Funding:** This study was funded by grants from the Academy of Finland (I.K.), the Sigrid Jusélius Foundation (I.K.), and the Norwegian Research Council (I.K). P.B.R. and J.E.M. are supported by the Francis Crick Institute, which receives its core funding from Cancer Research UK (FC001143, FC001178), the Wellcome Trust (FC001143, FC001119), and the UK Medical Research Council (FC001143, FC001119). The funders had no role in study design, data collection and analysis, decision to publish, or preparation of the manuscript.

**Competing interests:** The authors have declared that no competing interests exist.

phylum are *Plasmodium* spp. (the causative agents of malaria), *Toxoplasma gondii* (toxoplasmosis), and *Cryptosporidium* spp. (gastrointestinal and respiratory cryptosporidiosis), all life-threatening human pathogens. Apicomplexan parasites are dependent on an actomyosin based motor system that drives gliding motility that they use to invade and egress from cells of the host organisms [2,3]. The motor complex, called the glideosome, is located in a narrow space between the parasite plasma membrane and an inner membrane complex (IMC), which is structurally unique to this class of parasites [2–4].

Myosins are a large and diverse family of motor proteins, with >30 classes, found across all eukaryotic organisms [5]. Although the myosin classes are functionally and structurally distinct, they share several conserved domains [6,7] and produce force and movement *via* the same basic mechanism, making cyclical interactions with actin, coupled to the break-down of ATP to ADP and inorganic phosphate ($P_i$). Most myosins characterized to date move towards the plus-end of actin filaments, with the exception of the reverse-directed, class VI myosins. The myosin heavy chain consists of an N-terminal motor domain that is well-conserved and contains the catalytic site and actin binding site, which attaches and releases actin during the ATPase-cycle. The motor domain is followed by a "neck" region, which binds one or more calmodulin-like light chains. This region is functionally important because structural rearrangements within the motor domain cause it to rotate through a large angle (>60˚), and it acts as a lever arm to amplify the motion, producing the active "power stroke". Class VI myosins have a unique insert within the neck sequence that reverses the power stroke direction. Following the neck region, most myosins have an extended, C-terminal "tail", which is highly diverse in both sequence and structure and is responsible for targeting the motor to its cellular cargo. Some myosins have a coiled-coil forming region within the tail that causes the heavy chains to dimerize [8].

*Plasmodium* spp. have six myosin genes: three parasite-specific class XIV myosins, two reverse-directed class VI myosins, and one class XXII myosin [9]. One of the class XIV myosins, *Pf*MyoA, is essential for gliding motility and is the best-characterized of the parasite myosins. It is unusually small, consisting of just the canonical motor domain and neck region that bears two light chains: essential light chain (*Pf*ELC) and myosin tail interacting protein (*Pf*MTIP) [10,11]. As *Pf*MyoA completely lacks a tail region, it is thought to be unable to dimerize on its own. *Pf*MyoA is anchored via MTIP to the IMC by binding two lipid-modified glideosome-associated proteins (*Pf*GAP45 and *Pf*GAP50) [12]. To generate gliding motility, *Pf*MyoA interacts with *Pf*Act1 filaments [13–15] that are linked to the plasma membrane *via* the glideosome-associated connector (GAC; [16]), which binds to transmembrane adhesins of the thrombospondin-related anonymous protein family [17]. During gliding, *Pf*MyoA moves along actin, pulling the IMC towards the anterior (front end) of the parasite, while pushing the actin filaments and associated plasma membrane, rearwards [18]. X-ray crystal structures have been determined for the truncated motor domains of *Pf*MyoA ortholog from *T. gondii* [19], *P. falciparum* MyoA [20], full-length *Pf*MyoA decorated with *Pf*ELC and *Pf*MTIP [21], trimeric complexes of the tail segment and light chains: *Pf*ELC and *Pf*MTIP [22], and finally a cryo-EM structure of motor domain decorated actin [23]. For brevity, herein we refer to the *P. falciparum* proteins without the species-specific prefix "*Pf*".

Apicomplexan actins differ from the well-characterized canonical actins from yeasts, plants, and animals in some important aspects. While canonical actins form filaments, which can be micrometers to tens of micrometers in length, apicomplexan actin filaments in similar *in vitro* conditions are short, typically around 100 nm [14,15,24]. Nevertheless, they contain intrinsic capacity to form tens of micrometer long filaments in high micromolar concentrations *in vitro* [25]. In malaria parasites, visualization of filamentous actin has been elusive, and recently work with actin-chromobodies visualized actin networks at different development stages [26].

For *T*. *gondii* actin, an isodesmic polymerization mechanism was suggested to be the reason behind the short filament length [27]. However, we have shown that the polymerization pathway and kinetics of Act1 are very similar to canonical actins [24], and the short length of the filaments is likely caused by a high fragmentation rate [28]. In a recent study based on TIRF microscopy, it was suggested that the critical concentration for polymerization of Act1 is at least an order of magnitude higher than that of canonical actins [25]. Also the link between ATP hydrolysis and polymerization seems to be different in apicomplexan compared to canonical actins [15,24,28]. Whereas canonical actins polymerize preferably in the ATP form and only catalyze ATP hydrolysis in the filamentous (F) form, ATP hydrolysis in the monomer causes *Plasmodium* actins to form short oligomers [15]. These observations indicate that there may be differences in the catalytic mechanism of ATP hydrolysis and in the activation of polymerization between the apicomplexan and higher eukaryotic actins.

Malaria parasites go through several transformations between very different cell types during their life cycle, and some of these forms show rapid and highly directed movement. Merozoites penetrate red blood cells in seconds, the midgut-penetrating ookinetes can move at ~5 μm/min [29] and the mosquito-transmitted sporozoites move with an impressive average speeds of 1–2 μm/s for tens of minutes [30]. Recently, it was proposed that the speed and force generation of MyoA are fine-tuned by phosphorylation of Ser19 in the N-terminal helix [20]. In addition, the light chains are required for maximal speed as measured by *in vitro* actin-gliding assays [10,11]. The spatial relationship between the full-length MyoA:ELC:MTIP complex and its binding partner, filamentous actin, is necessary in order to understand the role of the light chains in determining force, speed, and step size of the motor complex as well as its association with the glideosome complex.

Here, we report the structure of full-length MyoA, with its ELC and MTIP light chains, bound to the Act1 filament and a new high-resolution structure of the non-decorated Act1 filament. Our high-resolution structures show interactions at the *Pf*Act1:*Pf*MyoA binding interface and the positioning of the two light chains (ELC and MTIP) on the MyoA neck region. Additionally, we characterize the common mode of binding to actin for different classes of myosin despite divergent amino acids at the binding interface.

## Results and discussion

### High-resolution structures of malaria parasite actomyosin and actin filaments

We reconstituted the malaria parasite actomyosin filament in the rigor state *in vitro* using recombinantly expressed Act1 and MyoA:ELC:MTIP complex in the presence of a small cyclic peptide, jasplakinolide, which stabilizes actin filaments and has been used for cryo-EM studies on Act1 filaments previously [13,15,23]. After plunge-freezing and cryo-EM imaging, we observed three types of filaments: bare Act1 filaments and filaments either partially or fully decorated by the MyoA:ELC:MTIP complex. By picking well-decorated filaments from the micrographs (**S1 Fig**), we determined the malaria parasite actomyosin structure using helical reconstruction methods [31] to an overall resolution of 3.1 Å, as estimated by Fourier shell correlation with the 0.143 threshold criterion (**Figs 1 and S1 and Table 1**).

For direct comparison, we also determined the structure of the non-decorated Act1 filament at an average resolution of 2.6 Å (**Figs 1 and S2 and Table 1**). Visual inspection of the density maps of both complexes agrees with the global resolution estimations (**Figs 1 and S3– S5 and S1–S3 Movies**). Based on the maps, we built atomic models for an actomyosin filament consisting of four Act1 molecules and two MyoA molecules and for the non-decorated Act1 filament consisting of five actin subunits (**S5 Fig**).

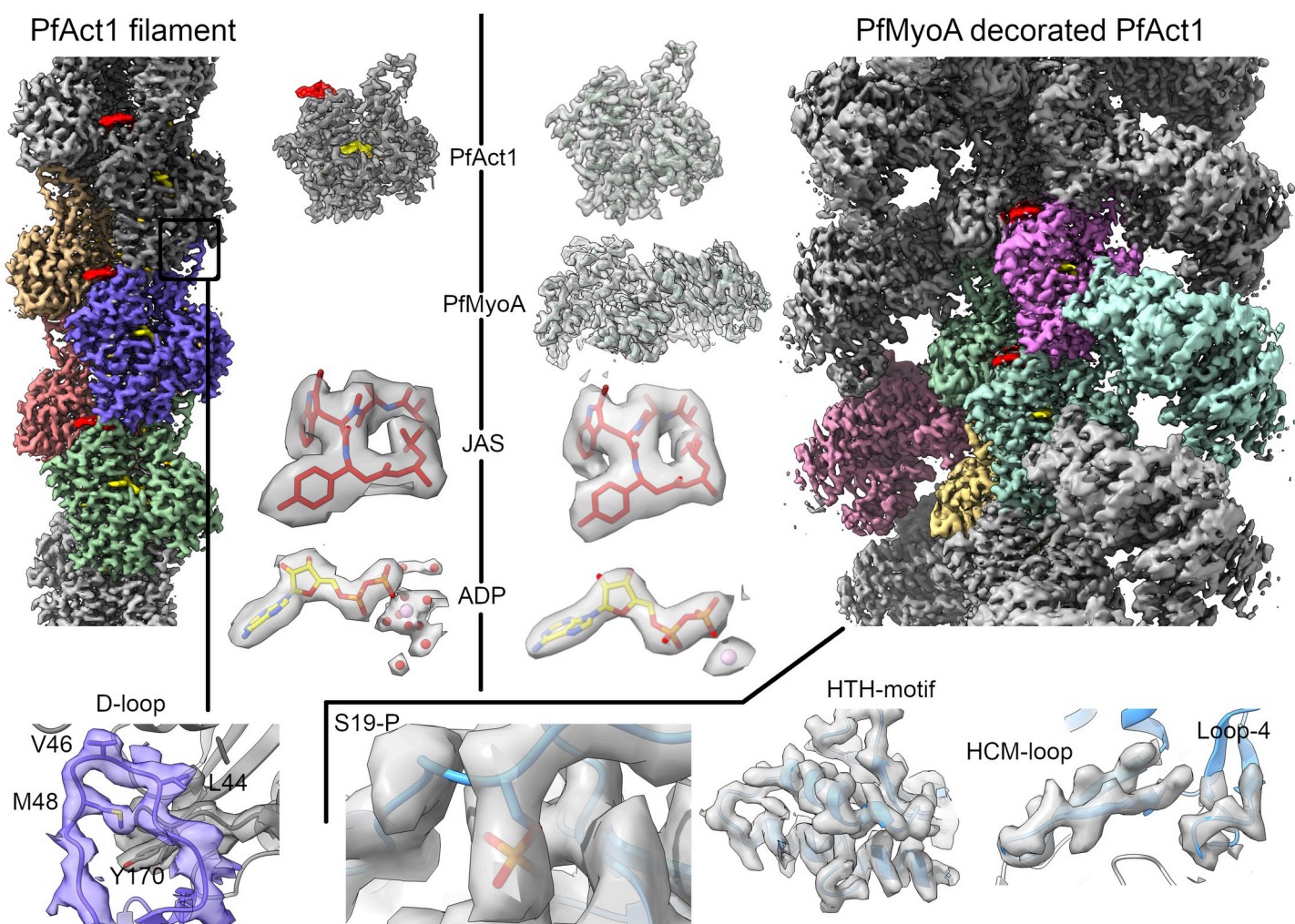

**Fig 1. High-resolution reconstructions of filamentous Act1 (left) and MyoA-decorated Act1 (right).** In the density maps, the central individual actin protomers and myosins are depicted in different colors. Jasplakinolide (JAS) and ADP are shown in red and yellow, respectively. The top middle part shows density maps around individual protein molecules and small molecule ligands as labeled. The bottom row shows density maps of selected structural motifs, as labeled, within the interfaces in the complexes. Additionally, density around phosphorylated Ser19 (S19-P) is shown.

As typical for actomyosin cryo-EM structures [23,32–35], the resolution of the Act1:MyoA: ELC:MTIP structure decays radially away from the helical axis, having a well-resolved core, consisting of the actin filament and the myosin motor domain, whereas density for the MyoA neck region is weaker, but nevertheless allowed us to locate density for the light chains, ELC and MTIP (**Fig 2**). The actin core in both of our structures is particularly well-resolved compared to other actin filament structures (**S3 and S4** Figs). Thus, the molecular details of F-actin, including the small molecules ADP and jasplakinolide as well as the active site metal and water molecules, can be visualized in greater detail (**Fig 1**) than in myosin-decorated or undecorated actin filament structures [13,32–41]. The majority of the MyoA motor domain is well-ordered, and side chains can be placed into density, especially within the actomyosin interface, allowing analysis of specific contacts between MyoA and Act1 in detail.

ELC and MTIP bind to the MyoA neck in a compact manner, presumably stiffening the neck region so it acts as a relatively rigid lever arm (**Fig 2**), consistent with the requirement of both light chains for maximal actin gliding speed as measured in *in vitro* motility assays

**Table 1. Data collection and refinement statistics.**

| *Data collection* | Act1-MyoA | Act1 |
|---|---|---|
| Magnification | 75000 x | 75000 x |
| Defocus range (μm) | 0.8–2.6 | 0.8–2.6 |
| Voltage (kV) | 300 | 300 |
| Microscope | Titan Krios | Titan Krios |
| Detector | Falcon 3 | Falcon 3 |
| No. of frames | 46 | 46 |
| Pixel size (Å/pixel) | 1.09 | 1.09 |
| Electron dose (e⁻/Å²) | 49.22 | 52.4 |
| No. of micrographs | 2786 | 3953 |
| *Reconstruction (Relion)* | | |
| Software | Relion 2.1/3beta | Relion 3.0.7 |
| Segments | 239225 | 305480 |
| Box size (px) | 512 | 328 |
| Rise (Å) | 28.34 | 28.37 |
| Azimuthal rotation (°) | -168.48 | -167.65 |
| Average resolution (Å) (FSC = 0.143) | 3.1 | 2.6 |
| Model resolution (Å) (FSC = 0.5) | 3.2 | 2.7 |
| Map sharpening B-factor (Å²) | 104 | 67 |
| *Model building (Phenix)* | | |
| No. of atoms | 24052 | 15690 |
| No. of amino acid residues | 3020 | 1860 |
| No. of water atoms | | 140 |
| No. of ligand atoms (Mg²⁺, ADP, JAS) | 12 | 15 |
| Average B-factor (Å²) | 62 | 37 |
| Average B-factor for ligand atoms (Å²) | 42 | 36 |
| Average B-factor for water (Å²) | | 31 |
| R.m.s.d. bond lengths (Å) | 0.006 | 0.003 |
| R.m.s.d. bond angles (°) | 0.587 | 0.661 |
| CC volume | 0.84 | 0.83 |
| CC masked | 0.88 | 0.87 |
| Molprobity score | 1.60 | 1.52 |
| Clash score | 6.08 | 3.51 |
| Ramachandran plot favoured/allowed/outliers (%) | 96.7/3.3/0 | 98.5/1.5/0 |
| *Deposition codes* | | |
| PDB | 6TU7 | 6TU4 |
| EMDB | EMD-10590 | EMD-10587 |

[10,11]. To confirm that our recombinant MyoA:ELC:MTIP complex was fully-functional we measured its ATP-driven power stroke, using single-molecule optical trapping nanometry [42] (see Materials and Methods). The average power-stroke, determined by histogramming the amplitude of many individual actomyosin binding events, was 4.5 +/- 0.4 nm s.e.m. (1535 events from 8 replicates, **Fig 2C, upper**). The distribution of event lifetimes was monotonic and controlled by the rate of ATP binding to the rigor complex (**Fig 2C, middle**). Ensemble-averaging of events [43] revealed the power stroke was biphasic, comprising a rapid initial movement of ~4 nm followed by a subsequent movement of ~1.5 nm (**Fig 2C, lower**). Previous studies, indicate the biphasic nature of the power stroke arises from structural changes associated first with phosphate and later with ADP release [44].

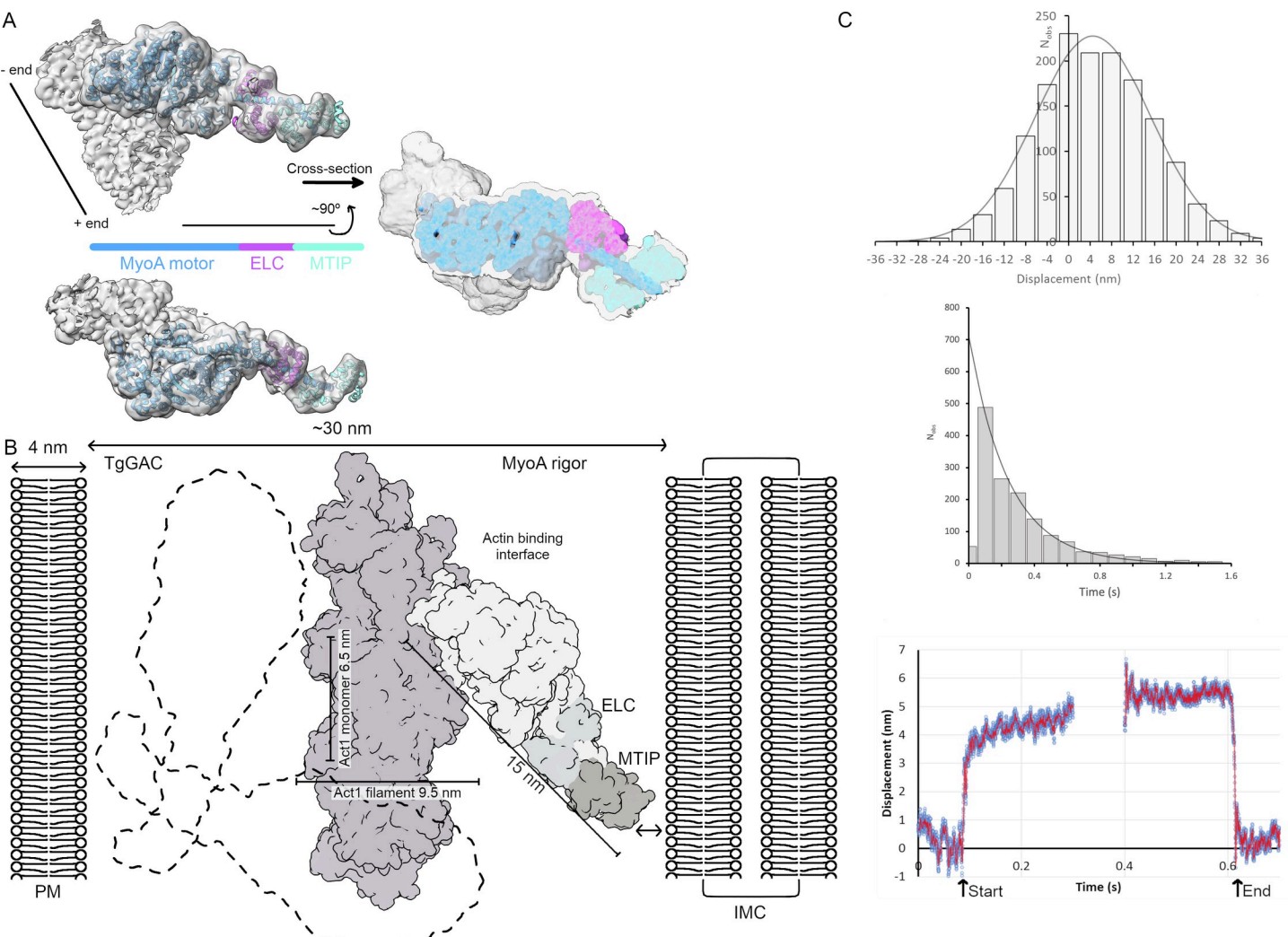

**Fig 2. A model of the light chains ELC and MTIP bound to MyoA.** (A) An unsharpened map (gray) filtered with LAFTER shows the location of the MyoA motor (blue) and the light chains ELC (magenta) and MTIP (cyan). (B) The Act1:MyoA:ELC:MTIP model in the rigor state shown in the context of the membrane-delimited sub-pellicular compartment. An outline of GAC (dotted lines) in two randomly chosen orientations with approximate dimensions derived from the *T. gondii* GAC small-angle X-ray scattering (SAXS) low-resolution envelope [16]. (C) Results of single molecule optical tweezer assays are presented. (Upper) A displacement histogram for MyoA: ELC:MTIP at 2.5 μM ATP. Power stroke was determined as the shift in the Gaussian peak from 0 nm (gray line), which has an average ~4.5 nm. (middle) An attached lifetime histogram for muscle actin-MyoA:ELC:MTIP events measured at 2.5 μM ATP. The detachment rate (grey line) was determined by fitting a single-exponential to the histogram. The event lifetime distribution (middle panel) fitted well to a mono-exponential decay (least squares fit gives a rate constant 4.1 s$^{-1}$) consistent with detachment being controlled by ATP binding to the rigor complex (at 2.5 uM ATP was $1.4 \times 10^6$ M$^{-1}$ s$^{-1}$). (Lower) The panel shows the ensemble average of 598 binding events. Each event was synchronized to its start [arrow] and end point [arrow] defined by the change in system stiffness, determined from the amplitude of a 200 Hz sinusoidal forcing function applied by the optical tweezers. The synchronized data were averaged. The average displacement at the start of the event (~4 nm) is smaller than that at the end of the event (~5.5 nm). This observation implies the power stroke is generated in two phases, ~4 nm followed by a further motion of ~1.5 nm.

A consequence of the extended and rigid neck region is a space constraint for the actomyosin complex in the sub-pellicular compartment. The MyoA:ELC:MTIP complex would occupy approximately half the distance between the IMC and the plasma membrane (~30 nm; **Fig 2B**), which is similar to the spacing between thick and thin filaments in vertebrate muscle sarcomeres [45]. In the parasite, when the additional space taken up by the actin filament (diameter ~6.5 nm) and GAC (D$_{max}$ in solution ~25 nm; [16]) is considered, it is clear that there must be significant geometric and steric constraints on how the actomyosin complex might be arranged (**Fig 2A and 2B**). It is possible that the attachment of MyoA *via* its MTIP light chain

to the GAP45:GAP50 complex in the IMC might allow azimuthal movement so the motor domain might approach the actin helix from different angles. Also the structure of GAC may be different in the tightly packed pellicle than in solution. These structural considerations will need to be revisited when the 3D geometry of the full glideosome complex is understood.

The MyoA motor domain has a conserved overall structure with several known subdomains common to functional myosin motors (S6 Fig). The upper and lower 50 kDa subdomains are separated by the actin-binding cleft, which is in a closed conformation in our complex (Fig 3). The central transducer region consists of seven β strands (S6 Fig). The lower 50 kDa subdomain contains a helix-turn-helix (HTH) motif, involved in intimate contacts with Act1, while the rest of the actin-contacting residues are in the hypertrophic cardiomyopathy (HCM) loop and in loops 2, 3, and 4 (Fig 3). The more distal parts, the converter and the relay helix, are involved directly in the conformational change upon the power stroke (Fig 2, [46]). Additionally, class XIV myosins have an N-terminal SH3 domain, the role of which is unclear but could be related to the regulation of myosin function during different stages of the

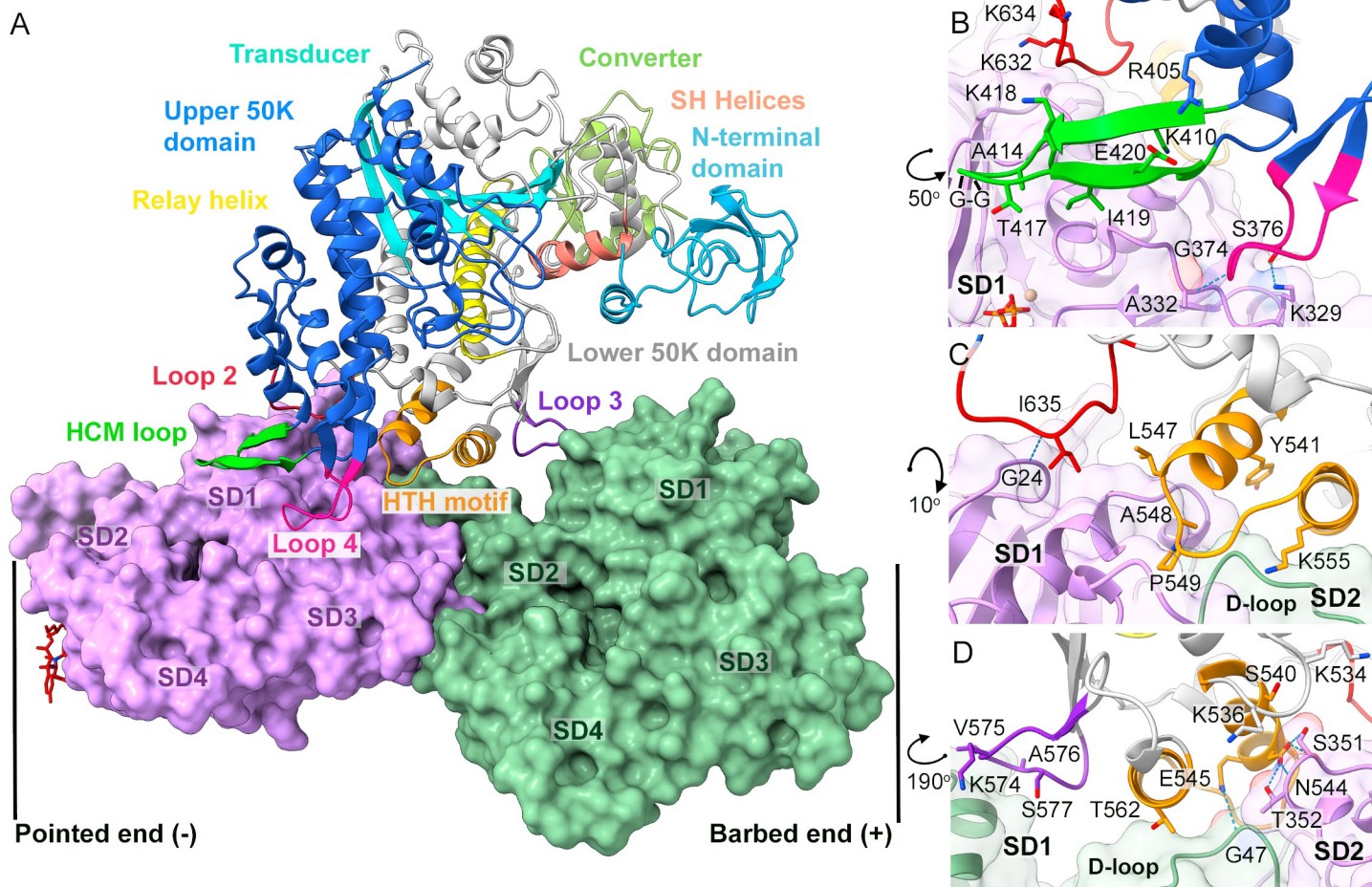

**Fig 3. Molecular interactions of MyoA with Act1.** (A) MyoA is shown as cartoon and Act1 as surface representation. The two adjacent actin protomers are colored with distinct colors. The MyoA motifs contributing to the interface as well as MyoA subdomains, and the N-terminal helix and SH3 domain are highlighted and labeled. (B) The HCM-loop lies on one actin protomer, contacting actin SD1. The basic loop 2 (Lys632 and Lys634) interacts with the acidic N-terminus of Act1. (C) The HTH motif in the lower 50K domain contributes most specific hydrogen bonds in the interface, interacting with the D-loop (SD2) and the C-terminal patch of Act1 between SD1 and SD3 in close vicinity of the D-loop in the adjacent actin subunit. (D) Loop 3 and the HTH motif visualized approximately 190° rotated with respect to panel A. Loop 3 (576–577) interacts with the adjacent actin monomer on SD1. The activation loop in the lower 50K domain protrudes towards the next SD1, with Lys534 interacting with the actin N-terminus.

parasite life cycle [47]. Our actomyosin rigor structure shows that neither the SH3 domain nor the N-terminal helix is in direct contact with the actin filament and the closest segments are about 18 Å away (**Fig 3**).

Overall, the MyoA structure in complex with actin is very similar (with an overall root mean square deviation (r.m.s.d.) for the $C_\alpha$ positions ~1.3 Å) to the rigor-like conformer found as one of the molecules in the asymmetric unit in the recently determined MyoA crystal structure [20]. However, there are several small but important changes in the complex. Notable differences allow interactions and avoid clashes with actin, including movement of the HCM loop (Ile409-Arg422) by nearly 4 Å, slightly smaller shifts in loop 4 (Glu369-Ala380), loop 3 (Pro572-Phe581), and the HTH motif (Val538-Asn563) as well as the ordering of loop 2 (Lys632-Ser639), which is not visible in the truncated MyoA crystal structure [20], but can be seen in the full-length complex with light chains [21]. In addition, there are several smaller rearrangements in more distal areas, in particular in the converter domain. The N-terminal helix, which is specific to apicomplexan MyoAs and accommodates the phosphorylated Ser19, is stacked between the SH3 domain and the motor domain, suggesting an important structural or functional role (**Figs 3 and S6; [20]**). The phosphorylation of Ser19 in the MyoA:MTIP: ELC complex is visible in the density map (**Fig 1**) and was confirmed using mass spectrometry, showing ~50% phosphorylation of the relevant peptide. In the crystal structure of *T. gondii MyoA*, the entire N-terminal helical extension is not visible in the electron density map [19] but is seen in most MyoA crystal structures [20,21] and MyoA:Act1 complex [23]. The MyoA: Act1 complex described by Robert-Paganin *et al.* (2021) has slightly different helical parameters, half a degree in twist and 1 Å smaller rise. The largest differences are in the HCM loop and at the tip of HLH with overall r.m.s.d. ~1.4 Å.

## Naked Act1 and myosin decorated Act1 filaments are nearly identical

We reconstructed the Act1 filament alone and obtained helical parameters that were similar to our Act1:MyoA complex (**Table 1**) with a similar rise and a 0.7˚ smaller azimuthal rotation for the complex. As previously noted for other actomyosin complexes [23,32,33,35], neither MyoA nor Act1 undergo large conformational changes when compared to the structures in isolation, and actin subunits superpose well over 372 residues (RMDS 0.34 Å). However, our highly-resolved density of the Act1 filament allows us to see most of the side chain orientations and details of ligand binding for ADP•$Mg^{2+}$ and jasplakinolide, as well as putative water molecules, including several in the vicinity of the active site (**Figs 1 and 5 and S4 Movie**). Active sites of filamentous Act1 or Act1:MyoA filament models are nearly identical, showing presence of ADP and its coordinated magnesium ion.

The Act1 filament has an internal cavity, which is extended towards the filament exterior (**Fig 4A and S4 Movie**). We modelled density in the cavity as a chain of water molecules (**Fig 5**), but we cannot exclude presence of other solute molecules. The end of the cavity is formed by the side chains of Asn116 and Trp80 (**Fig 4 and S4 Movie**). In monomeric Act1, the cavity is much shorter compared to the F-form (**Fig 4C**). In muscle F-actin, the cavity is limited by the bulkier Ile76 instead of Val77 in Act1 (**Fig 4**). Notably, residue 76/77 is an Ile in α- and γ-actins, but is also Val in β-actin as well as plant and yeast actins, which release phosphate faster than muscle actin [48,49]. We have previously noted the importance of Gly115 substitution to alanine in Act1; alanine at this position slows down phosphate release compared to glycine, in particular in the Mg-bound form [15,28]. In skeletal or smooth muscle α-actin, Asn115 and Arg116 are sites of disease-associated mutations (N115S, N115T, R116Q, R116H) [50,51]. In yeast, mutations N115T and R116Q affect both nucleotide exchange as well as polymerization

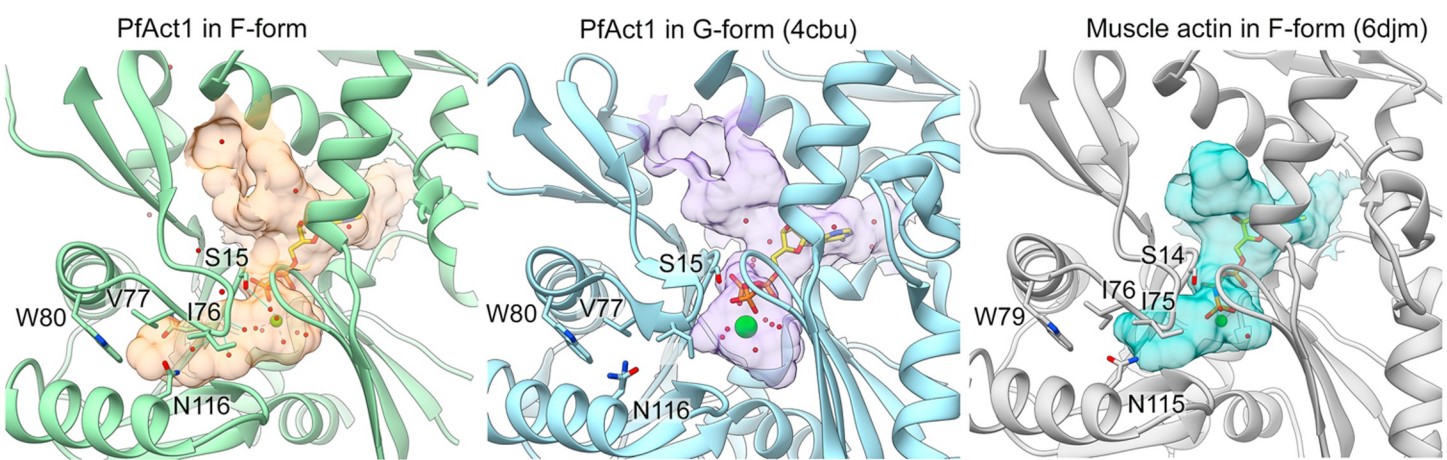

**Fig 4. Internal cavities in filamentous and monomeric actin structures.** The internal cavities were calculated using the CASTp server [74] and visualized as transparent surfaces using Chimera [66]. (left) The internal cavity in the Act1 filament and (middle) the corresponding channel in monomeric Act1 (4cbu; [15]). (right) The corresponding cavity in muscle actin (6djm; [38]) is shorter due to change from Ile76 to Val77 in Act1.

kinetics, in particular in the nucleation phase [52]. Therefore, these features of Act1 active site may be relevant to understand actins generally.

Based on crystal structures of monomeric actin, the side chain of Ser15 (14 in canonical actins) has been established as a nucleotide sensor, which is turned away from bound ATP but hydrogen bonded to the β-phosphate of ADP [53]. In our structure of the ADP-bound filamentous Act1, the side chain of Ser15 points away from the nucleotide, not contacting the ADP phosphates, thus more closely resembling the conformation in crystal structures of the ATP-bound state (**Fig 5**). Instead, the Ser15 main chain NH group interacts with the β-phosphate.

## Actomyosin interface has conserved features

The MyoA domain architecture is conserved, as in other myosins, regardless of the myosin class (**Figs 2 and 3**), but residues in the actin-binding interface are generally not conserved (**S6 and S7 Figs**). The Act1 surface is relatively flat, and thus MyoA lacks large protrusions, which would penetrate deep into the filament pockets (**Fig 3A**). The interface is mainly formed by contacts between the HTH motif, the HCM loop, and loops 2, 3, and 4 (**Fig 3**) similarly to previous structures [23,34,35,46]. We analyzed several actomyosin filament structures in the rigor conformation (**S7 Fig**) by structure-based sequence alignment (DALI, [54]) of the motor domains and compared the fractional buried surface area (PDBePISA, [55]) of MyoA residues in the actin contacting segments. Although, most interacting amino acids are not conserved among different myosin classes, especially in loops 2,3 and 4 (**S6 Fig**), they have a strikingly conserved pattern of buried surface area within the actin interface (horizontal blue bars in **S7 Fig**). Interestingly, when we set a threshold for interface residues that are >25% buried in all six myosins, a common interface consists of nine residues (long horizontal red boxes; Thr412, Ala414, Ile419, Gly421, Asp544, Leu547, Ala548, Pro550, Ile635) mostly contacts from HCM-loop and HLH substructures.

Recently, Robert-Paganin *et al*. (2021) defined conserved (15 residues) and ancillary (9 residues) interface in MyoA (green and cyan boxes in **S7 Fig**, respectively). Their conserved interface partially overlaps with our analysis, but our core interface consists of 9 residues. Additionally, we noted that each respective myosin motor makes extensive contacts with actin that are not present in the other myosins (gray boxes and yellow-highlighting in **S7 Fig**), which we summarize here. Within loop 3, Myo 1b has extensive contacts with actin that are

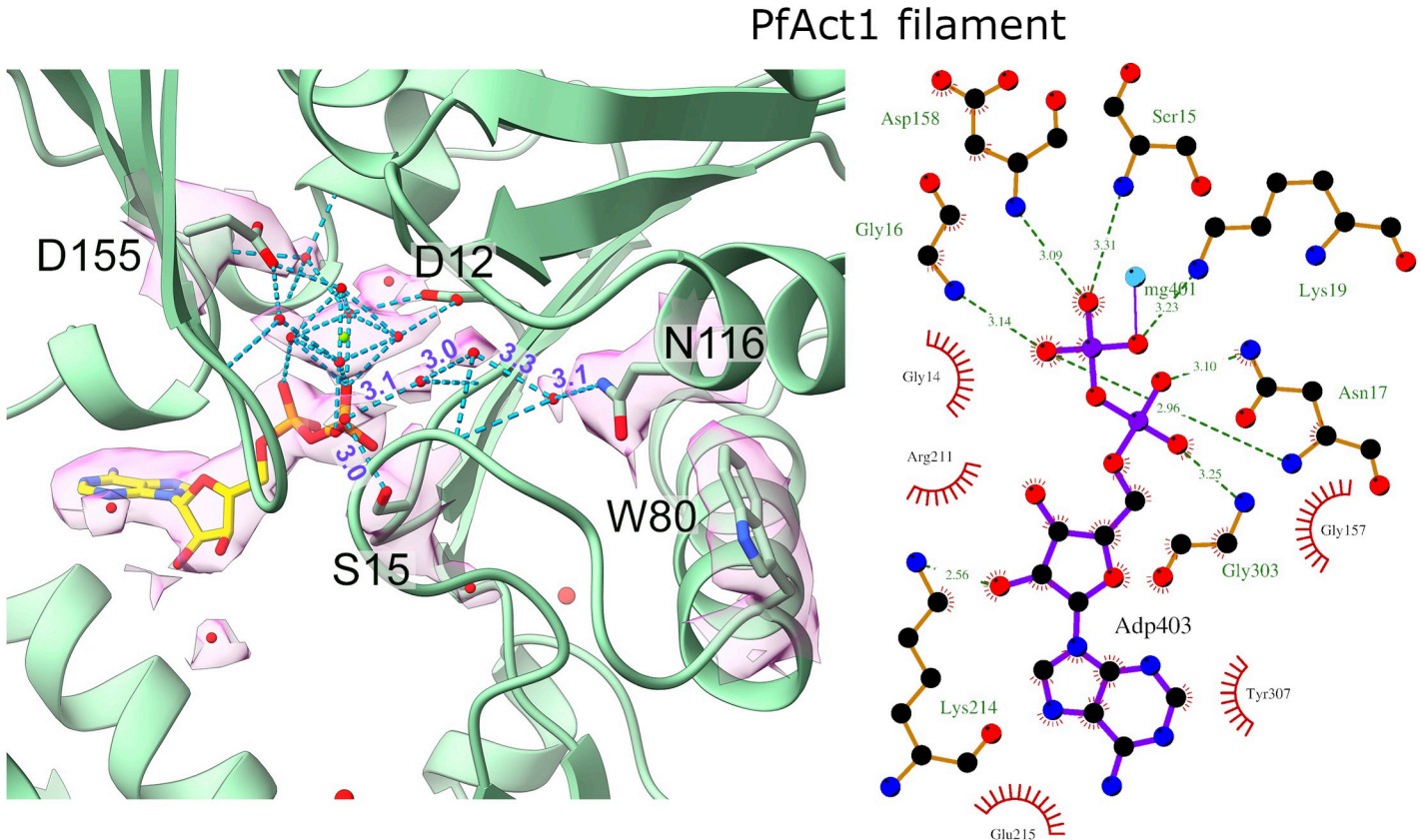

**Fig 5. Details of the nucleotide binding site in the Act1 filament.** (left) Density map around the nucleotide-binding site, including ADP (sticks), Mg$^{2+}$ (green sphere) with coordinating water molecules and putative water molecules (red spheres) in the internal cavity. (right) Schematic presentation of ADP interactions in the filamentous Act1 active site.

not present in other myosins. In loop 4, MyoA uses it to interface actin more (yellow-shaded box) similarly to Myo 6. At the end of loop 2, Myo 7 has an additional patch of non-charged residues in addition to a charged-patch within loop 4. Within the activation loop of the HLH motif, Myo 2c has more interfacing residues; we noted double proline, and three additional residues at loop 2. The HCM-loop is 5 residues longer in Myo 6 than others, and the tip of loop makes extensive contacts with actin.

## Helix-Turn-Helix (HTH) interactions

In MyoA, the majority of specific contacts are formed by the HTH motif in the lower 50K domain, which contacts the actin-actin interface (buried surface area 554 Å$^2$), where the D-loop of one actin inserts into the cleft between subdomains 1 and 3 of the adjacent actin (**Fig 3B and 3C**). The first half of the HTH motif (Ser540-Gly550) forms extensive contacts with both actin subunits, whereas the latter part (Gly551- Lys565) interacts mostly with the D-loop. The Asp544 side chain of MyoA forms hydrogen bonds with the backbone amine of Thr352 in Act1 and the side-chain or backbone amine of Ser351. Asp544 also interacts *via* its backbone carbonyl with the backbone amine of Ala548 in MyoA. This interaction network between Act1 and MyoA occurs just before the kink in the relay helix, possibly stabilizing it. The tip of the HTH motif (Ala548-Pro549-Gly550) inserts into a hydrophobic cleft in Act1, lined by Ile347-Leu352 and Tyr145-Thr150 as well as the D-loop residues Met46-Val47 from the adjacent actin subunit.

## Hypertrophic Cardio-Myopathy and other surface loop interactions

The HCM loop (Fig 3B) forms a second important interface to Act1 (buried surface area 284 $Å^2$) that is complementary in shape, and we detect no hydrogen bonds at the interface. Additional contact points are created by loop 2, which is in close proximity to the HCM loop. The backbone atoms of Ile635 and Gly633, hydrogen bond the actin backbone atoms Gly24 and Asp26 respectively and the Ile635 side-chain inserts into a small patch on actin lined by Gly25, Asp26, Ser345, Ile346, and Ser349. The basic loop2 residues; Arg606 (in the strut segment), Lys634, and Lys637 coordinate actin Asp26, Asp25, and Ser349/Ser351, respectively, with distances ranging between 3.8 and 4.5 Å. The TEDS phosphorylation site [56] Thr417, located at the tip of the HCM-loop, is solvent accessible, and the density map suggests that it is not phosphorylated. Interestingly, our structural study of the actomosin complex did not require phosphomimetic mutation Thr417Asp in contrast to studies of the truncated MyoA [23], which results in a shift of the position of the HCM loop with respect to MyoA, and where the aspartic acid contributes buried surface area to the interaction.

Loop 3 has a small interaction point with Act1 sub-domain I, mediated by Val575/Ala576 in MyoA and Tyr92/Asn93 in actin. Finally, Loop 4 residues, Ala373-Asp377, have extensive interactions with Act1 throughout Lys329 to Glu335.

## Concluding remarks

The malaria parasite actomyosin filament structure reported here will serve as a foundation for understanding gliding motility in Apicomplexa as well as force generation in a variety of actomyosin motor systems. We visualize the actin-myosin interface at high-resolution and also present the structure of the entire core motor complex including the light chains, which are essential for fast motility. In addition, we show that the actomyosin light chain complex produces a power stroke similar to other myosins with two light chains [57]. The extremely well-resolved bare Act1 filament assigns accurate side-chain conformations and reveals new active site features. The structures describe several protein-protein interfaces and will be important tools for evaluating the actomyosin motor as a drug target.

## Materials and methods

### Optical trap assay

All chemical reagents were obtained from Sigma unless stated otherwise. Experiments were performed using a custom-built optical trapping instrument based around an inverted, Zeiss Axiovert 135 microscope, that has been described previously [44]. Briefly, microscope flow-cells were prepared from a microscope slide and coverslip fixed together with double-sided adhesive tape leaving a flow-channel of ~30 μl volume. Coverslips were previously coated with 2.4 μm diameter glass microspheres diluted into a 0.1% nitrocellulose solution. MyoA:ELC: MTIP was infused into the flow-cell at 1 μg ml$^{-1}$ in an assay buffer "AB-" containing 25 mM KCl, 25 mM imidazole, 4 mM MgCl$_2$, and 1 mM EGTA, pH 7.4, at ~23°C. Flow-cell surfaces were then passivated using four chamber volumes of AB− with 1 mg ml$^{-1}$ BSA. The flow-cell was then infused with "AB+" buffer comprising AB- supplemented 0.3 μM ATP, 2 mM creatine phosphate, 0.1 mg ml$^{-1}$ creatine kinase, 50 mM DTT, 3 mM glucose, 100 μM glucose oxidase, and 20 μM catalase, 0.2 nM rhodamine-phalloidin labeled, 10% biotinylated, filamentous porcine actin and neutravidin-coated 1μm diameter polystyrene beads. Actin filaments were visualised using 532 nm laser excitation. The incoming 532 nm laser was reflected by a dual-dichroic mirror (532 nm and 1064 nm, Semrock custom dichroic) which also reflected the near infra-red (1064nm) optical trapping laser onto the specimen. Rhodamine fluorescence

from the labelled actin was imaged via a 580-640nm bandpass filter using a custom-built, image-intensified CMOS camera. The polystyrene beads were visualised using bright-field optics which allowed a single actin filament (5–7 μm long) to be captured at either end between two beads held in two, independently controlled, optical traps. The so-called "dumb-bell" of bead-actin-bead was then held taut and positioned in close proximity to one of the surface-immobilized MyoA:ELC:MTIP coated beads. Under conditions used here only about 1-in-3 of the surface beads produced actomyosin interactions due to the low surface-density of MyoA molecules. Bead positions were measured by projecting their bright-field images onto two, 4-quadrant, photodiode detectors (bandwidth >10kHz, resolution <1nm). Digital data was collected at 10kHz and saved to computer. Optical trap stiffness was 0.02 pN nm$^{-1}$ and thermal noise, governed by the equipartition principle, was ~15 nm r.m.s. When actin bound the surface-immobilized MyoA positional noise fell dramatically because system stiffness increased to >0.25 pN nm$^{-1}$ as the actomyosin bond stiffness acts in parallel to the trap stiffness. To increase the time-resolution of our measurements the optical tweezers were moved sinusoidally (50 nm peak-to-peak, 200 Hz) producing a ~12 nm p-p bead motion. Motion was quantified by computing a running discrete Fourier transform (software phase-locked-loop) of the bead motion. This enabled the start- and end-time of individual binding events to be identified with <10 ms jitter so that events could be synchronised in software and averaged to give an ensemble representation of the powerstroke time-course (see [43]).

## Preparation of proteins and cryo-EM grids

MyoA:ELC:MTIP and Act1 were expressed and purified as previously described [10,24]. For Act1 samples, ammonium acetate was removed by a spin column, and Act1 (13.1 μM) was polymerized in the presence of 13 μM jasplakinolide by adding 10X KMEI, in final 1x concentration of 10 mM Hepes, pH 7.5, 50 mM KCl, 4 mM MgCl2, 1 mM EGTA. Polymerized filamentous Act1 was diluted down to 0.13 μM, and treated with apyrase (77 μg/ml) for 20 min before addition of the MyoA:ELC:MTIP complex in a 1:1 ratio.

For Act1-only samples, no apyrase was used. After 30 min incubation, negatively stained or cryo-EM grids were prepared. The quality of the decorated filaments was first checked with negative staining, and after confirming the presence of decorated Act1 filaments, frozen-hydrated samples were prepared on Quantifoil 2/2, 200 mesh in-air glow-discharged grids using an Mk III Vitrobot (FEI). 3 μl of sample was applied on a grid and blotted for 3 s before plunge-freezing in liquid ethane.

## Act1:MyoA filament

Data consisting of 2815 movies were collected with a Titan Krios electron microscope equipped with a Falcon 3 camera, operated at 300 kV in counting mode. The magnification was 75 000x, corresponding to 1.09 Å pixel$^{-1}$, using a dose rate of 0.539 e$^-$/Å$^2$s$^{-1}$ with exposure time of 92 s for a stack of 46 frames. The dose per frame was 1.07 e$^-$/Å$^2$, and we used defoci from 0.8 to 2.6 μm. The movies were aligned with MotionCor2 [58] within the Scipion framework [59]. Contrast transfer function parameters were estimated by CTFFind 4.1.5 [60], and the resulting 2786 good images were used for further data processing. To confirm data processability, we picked decorated filaments by hand and performed low-resolution reconstructions from 2x binned images of a few hundred particles using symmetry parameters defined for Act1 [13] or refined helical symmetry parameters in Relion 2.1 [31,61] using a featureless cylinder as a starting reference. The reconstruction converged to a filamentous structure with protruding densities typical for actomyosin structures. Further inspection revealed that homologous myosin motor domains [34] as well as a filamentous Act1 model (PDB code:

5ogw [13]) fit the density well. Decorated filaments were iteratively picked with Relion autopick using reference-free Relion 2D classes as templates with an inter-box distance of two subunits with a rise of 28 Å. The resulting 381 256 particles were subjected to reference-free 2D classification in Relion, resulting in 239 225 fully decorated actomyosin particles. High-resolution volumes were generated in Relion 2.1 auto-refine using low-resolution models from 2x binned data as a reference. A soft mask was applied to the reference model, and reconstruction was continued from the last iteration. When the reconstruction converged, we continued refinement of the contrast transfer function and subsequent reconstructions with Relion 3 beta for two additional rounds. Refined half-maps were sharpened automatically [62], and global resolution was corrected for the effects of a mask [63] using the Relion postprocessing tool. We estimated local resolution with Blocres in Bsoft package version 1.8.6 with a Fourier shell correlation threshold of 0.3 [64,65].

### Act1 filament

We collected 3953 movies with a Titan Krios electron microscope fitted with a Falcon 3 camera in electron counting mode, operated at 300 kV. We used a nominal magnification of 75 000x, corresponding to 1.09 Å pixel$^{-1}$. Each movie was recorded as 46 frames with 1.17 e$^-$/Å$^2$ dose/frame using defoci from 0.8 to 2.6 μm. The fractionated movies were aligned globally in Relion 3.0.7. Contrast transfer function was estimated by CtfFind 4.1.10 from dose-weighted micrographs. Particles were picked automatically using four times the rise (28 Å). The resulting 336 552 particles were extracted and binned two times at 2.18 Å/pixel in a 328 pixel box before they were subjected to a reference free 2D classification, and resulted in 305 480 particles in the final data set. These particles were re-extracted as unbinned, and processed with Relion auto-refine with a reference mask and solvent flattening from a previous reconstruction (nominal resolution 4.4 Å) low pass filtered to 20 Å. After polishing and contrast transfer function refinement, the resulting map reached an average resolution of 2.6 Å, based on the 0.143 Fourier shell criterion. Visual inspection of the density map features agreed with the resolution estimate. After reconstruction, the map was masked and sharpened by the Relion postprocessing tool.

### Model building

**Act1:MyoA filament.** Act1 (PDB code 5ogw; [13]) was docked in density using Chimera [66], and we generated four additional symmetry copies. Two copies of the MyoA crystal structure in the rigor-state (PDB code 6i7d; [20]) were placed into the density using Chimera. We iteratively built a model using tools in ISOLDE 1.0b [67], Coot [68], and real-space optimized by Namdinator [69]. The local resolution of the reconstruction was estimated by Blocres [63], Chimera [66], and ChimeraX [70].

In addition, we note that the experimental map is of high quality. Before the MyoA crystal structure became available, we built a complete model of MyoA, *de novo*, based on a homology model of MyoA, the density map, and myosin models 1DFK and 6C1D [34,71] to guide manual model building. Our de novo MyoA model assigned the sequence register similarly to the X-ray crystal-structure (PDB code: 6i7d) and the last 25 amino acids were better resolved than in the X-ray structure. Additionally, phosphorylation of Ser19 was resolved with the correct sequence register.

### Act1 filament

Our previous model of the Act1 filament (PDB code: 5ogw; [13]) was placed in the density map in Chimera [66], and was manually built using model building tools in ISOLDE 1.0b3.

dev6 [67], Coot [68], and real-space refinement in Phenix 1.17 [72]. We built five copies of Act1, re-assigning side chain rotamers as required, and waters around Act1 monomers into density. Additionally, we built the active site with the $Mg^{2+}$-coordinated waters. Density for the $Mg^{2+}$-water complex was anisotropic, and the $Mg^{2+}$-coordinated waters were merged into $Mg^{2+}$-ion density. In order to keep ideal coordination of the water molecules, we created custom restraints for waters to restrain ideal distance and angles to the $Mg^{2+}$ ion.

## Low-resolution model of light-chain-decorated myosin A

Unfiltered 2x binned half-maps were filtered with LAFTER [73], and the MyoA model (6tu7) and a trimeric light chain model (6zn3; [22]) were placed into the filtered LAFTER map.

## Supporting information

**S1 Fig. Act1:MyoA filaments and resolution reconstruction.** (A) Representative micrographs of MyoA-decorated Act1 filaments. (B) Reference free classes derived from the micrographs. (C) Fourier shell correlation of the Act1:MyoA complex. Using the 0.143 Fourier shell threshold criterion, the global resolution is 3.1 Å. The masked curve was calculated from independently refined half-datasets with a soft-mask filtered to 15 Å.
(PDF)

**S2 Fig. Examples of the Act1 filaments and resolution reconstruction.** (A) Representative micrographs of Act1 filaments low-pass filtered at absolute frequency 0.15, (B) reference free classes derived from them, and (C) Fourier shell correlation of the Act1 filament. Using the 0.143 Fourier shell threshold criterion, the global resolution is 2.6 Å. The corrected curve was calculated from independently refined half-datasets with a soft-mask filtered to 15 Å in Relion.
(PDF)

**S3 Fig. Local resolution of the MyoA-decorated Act1 filament.** The local resolution estimation is based on the Fourier shell correlation threshold 0.3 calculated with Blocres in the Bsoft software package, which was displayed on the final sharpened map.
(PDF)

**S4 Fig. Local resolution of the Act1 filament.** The local resolution estimation is based on Fourier shell correlation threshold 0.3 calculated with Blocres in the Bsoft software package, displayed on the final sharpened map. The left panel shows a central section of the filament, and the right panel shows an individual actin protomer with the ligand densities highlighted.
(PDF)

**S5 Fig. Density maps of the Act1:MyoA complex and the individual sub-units.**
(PDF)

**S6 Fig. Structural domains and sequence comparison of the MyoA motor domain.** (Upper panel) The MyoA motor domain in two slightly different orientations to display the different functional motifs, shown in different colors and labeled. (Lower panel) In the structure-based sequence alignment, strictly conserved residues have red background, more than 70% identical residues are red in a blue frame. Amino acids in the actomyosin interface are indicated with cyan shading. Above the sequence, the actin-contacting structural motifs are indicated. The black bars under the sequence show fractional buried surface area for MyoA residues (1 bar <30%, 2 bars 30%-60%, 3 bars >60%). The sequence alignment is based on pairwise structural alignment against MyoA using the DALI server [Holm L. DALI and the persistence of protein shape. Protein Sci. 2020;29(1):128–40. Epub 2019/10/14. doi: 10.1002/pro.3749] [54], and further annotated using the ESPript 3.0 web interface [Robert X, Gouet P. Deciphering key

features in protein structures with the new ENDscript server. Nucleic Acids Res. 2014;42(Web Server issue):W320-4] [75].
(PDF)

**S7 Fig. Buried surface areas of different actomyosin complexes in the rigor conformation.** Actomyosin structures were analyzed using PDBePISA [Krissinel E, Henrick K. Inference of macromolecular assemblies from crystalline state. J Mol Biol. 2007;372(3):774–97] [55]. A structure-based sequence alignment for each myosin is shown in the vertical column. The fractional buried surface area is shown for each amino acid residue in the actin interface. Long horizontal red boxes show common interfaces (fractional BSA >25% in all six structures). Grey boxes highlight additional interactions, which are discussed in the main text. Green and magenta boxes around amino acid residues indicate conserved and ancillary, respectively, interfaces, as defined by Robert-Paganin *et* al. [23].
(PDF)

**S1 Movie. The Act1:MyoA map colored by local resolution.** First, the malaria parasite actomyosin complex is rotated around the helical axis to present the overall quality of the map used to build the actomyosin model. The latter part of the movie shows the maps for a longitudinal Act1 dimer and MyoA separately. The slider on the right indicates the color code for the resolution in different parts of the map from 2.5 (blue) to 4.5 Å (red).
(MOV)

**S2 Movie. Quality of the Act1:MyoA map.** First, the MyoA-decorated filament is rotated around the helical axis, demonstrating high-resolution electron density, in which the majority of side chains are visible, sufficient for *de novo* atomic model building. In the latter part of the movie, each of the central protein subunits (4 actins and 2 myosins) is depicted with a different color. Jasplakinolide, located between actin subunits, is shown in red and ADP in the cleft between the actin subdomains is shown in yellow.
(MOV)

**S3 Movie. High-resolution map of the Act1 filament colored by local resolution.** First, the filament is rotated around the helical axis, demonstrating high-resolution features. The latter part of the movie shows a single actin protomer. The slider on the right indicates the color code for the resolution in different parts of the map from 2.3 (magenta) to 2.9 Å (cyan).
(MP4)

**S4 Movie. A cross-section of Act1 showing the internal channel extending from the nucleotide binding pocket towards Asn116 and Trp80 on the surface.** The Act1 model is shown as surface representation, and the plane of view sections through the internal void revealing putative waters (red spheres) and ADP (sticks).
(MP4)

**S1 Data. Raw data related to the optical trapping experiments presented in Fig 2.**
(XLSX)

## Acknowledgments

We thank the Francis Crick Structural Biology Scientific Technology Platform for instrument access and computing support: Andrea Nans for help with data cryo-EM data collection, and Phil Walker, Andy Purkiss, and Andrea Nans for help with computing. We thank Donald Benton and Oliver Acton for helpful advice with image processing and Ulrich Bergmann from the Proteomics and Protein Analysis Core Facility at the Biocenter Oulu for mass spectrometry.

## Author Contributions

**Conceptualization:** Juha Vahokoski, Inari Kursula, Peter B. Rosenthal.

**Formal analysis:** Juha Vahokoski, Justin E. Molloy, Inari Kursula, Peter B. Rosenthal.

**Funding acquisition:** Justin E. Molloy, Inari Kursula, Peter B. Rosenthal.

**Investigation:** Juha Vahokoski, Lesley J. Calder, Andrea J. Lopez, Justin E. Molloy, Inari Kursula, Peter B. Rosenthal.

**Methodology:** Juha Vahokoski, Lesley J. Calder, Andrea J. Lopez, Justin E. Molloy, Peter B. Rosenthal.

**Project administration:** Inari Kursula, Peter B. Rosenthal.

**Resources:** Justin E. Molloy, Inari Kursula, Peter B. Rosenthal.

**Supervision:** Justin E. Molloy, Inari Kursula, Peter B. Rosenthal.

**Validation:** Juha Vahokoski, Inari Kursula, Peter B. Rosenthal.

**Writing – original draft:** Juha Vahokoski, Inari Kursula.

**Writing – review & editing:** Juha Vahokoski, Justin E. Molloy, Inari Kursula, Peter B. Rosenthal.

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
