## [Decision Letter · Decision Letter 0]

17 Oct 2021

Dear Dr. Kursula,

Thank you very much for submitting your manuscript "High-resolution structures of malaria parasite actomyosin and actin filaments" for consideration at PLOS Pathogens. As with all papers reviewed by the journal, your manuscript was reviewed by members of the editorial board and by several independent reviewers. In light of the reviews (below this email), we would like to invite the resubmission of a significantly-revised version that takes into account the reviewers' comments.

Overall the reviewers are supportive of the work but raised some important issues. Rev #1 and #2 are deploring the absence of in vitro or in vivo data. This  limitation could  be addressed by exploring experimentally the importance of the identified pocket. Importantly also Rev #3  requests that a number of inconsistencies, lack of clarify or  missing details to be fixed.

We cannot make any decision about publication until we have seen the revised manuscript and your response to the reviewers' comments. Your revised manuscript is also likely to be sent to reviewers for further evaluation.

Sincerely,

Dominique Soldati-Favre

Section Editor

PLOS Pathogens

Kasturi Haldar

Editor-in-Chief

PLOS Pathogens

orcid.org/0000-0001-5065-158X

Michael Malim

Editor-in-Chief

PLOS Pathogens

orcid.org/0000-0002-7699-2064

Reviewer's Responses to Questions

**Part I - Summary**

Reviewer #1: The focus of this paper is the use of electron microscopy in the structure determination of the actomyosin motor complex from Plasmodium falciparum. The complex includes the parasite actin filament decorated with the class XIV myosin motor and two associated light-chains.

The structures provide genuine new insight into variety of aspects: i) the myosin-actin interface in atomic detail, ii) the light-chain binding region, including the essential light chain and the myosin tail interacting protein and iii) the arrangement of the actomyosin motor complex in the sub-pellicular compartment.

Slightly disappointing that there were no biological experiments to probe the structural observations at the myosin-actin and light chain interfaces. If any updates on this are available or feasible within a reasonable timeframe it would enhance the paper. Despite this, and although this paper is a pure structure biology study, it is a significant step forward in our understanding of the actomyosin complex and is therefore suitable for publication in Plos Path.

Reviewer #2: The manuscript from Vahokoski et al. describes the 3.1 angstroem resolution structure of plasmodium actin filament decoreated with MyoA motors. PfMyoA is essential for parasite entry into red blood cells and as such represent an appealing drug candidate. The manuscript also describes the higher resolution of a bare plasmodium actin filament. The manuscript is well written with detailed atomic description of the structures. The data nicely show that plasmodium MyoA-actin interaction shares highly similar structure with other actomyosin structures, although there is large divergence between plasmodium and mammalian myosin motor sequences. Additionally, the structure of the myosin-actin complex is almost superimposable with the isolated structures of either PfAct1 filaments or of the MyoA protein. The high resolution structure of plasmodium actin filament identified a unique small cavity absent in other actin structures that could accommodate a non-identified solute.

Overall the results are highly similar to what was published in March 2021 (Robert-Paganin J. et al. Nat. Commun) and do not bring much additional information. The current manuscript emphasizes that the two myosin light chains ELC1 and MLC1 are present in their CryoEM reconstruction, but the very low resolution of these proteins does not add details as to how these light chains arrange when myosin is bound to actin filaments.

The models described in this manuscript are compared to isolated structures of either myosins or actin filaments, but no comparison with the published decorated PfAct1-PfMyoA filaments is presented.

The authors also claim that they identified a novel pocket in actin filaments that is absent from other known F-actin structures available. However, no in vitro or in vivo experiment support their hypothesis that this pocket could accommodate a solute.

Reviewer #3: In this manuscript, Vahokoski and colleagues report a 2.6-A-resolution Cryo-EM structure of Plasmodium falciparum’s actin isoform Act1 stabilized with jasplakinolide. Additionally, they report the structure of Act1 decorated with Plasmodium’s myosin MyoA and two associated light chains at 3.1 A resolution. Parasites from the apicomplexa phylum – like plasmodium – use a specialised type of motility known as gliding, which requires this actomyosin complex as its driving motor. As such, the structures presented in this manuscript are clearly of high interest, as they are a good pharmacological target. The high resolution of both structures, and particularly of the naked actin filaments, represent a large technical advance over the available structures. The research presented in the paper is sound and I would, in general, recommend the paper for publications. However, I found that in its current version, the results – especially some of the figures – could be presented in a much clearer way.

**Part II – Major Issues: Key Experiments Required for Acceptance**

Reviewer #1: Slightly disappointing that there were no biological experiments to probe the structural observations at the myosin-actin and light chain interfaces. If any updates on this are available or feasible within a reasonable timeframe it would enhance the paper.

Reviewer #2: From the low-resolution structure of decorated filaments with MyoA:ELC:MTIP, the authors propose an arrangement within the parasite PM-IMC space suggesting that there is no space to accomodate GAC, which has a Dmax of about 25 nm. Since the authors have previously published a low resolution SAXS GAC model, it would be fair to discuss and model how such a glideosome structure could be formed in such a confined space. The current Fig. 2 should be expanded to also contain GAC.

No in vitro or in vivo experiments were done to support structural findings. The authors claim that there is a specific cavity within PfAct1 filaments that is absent from other known structures but they have not validated experimentally the importance of this pocket. Mutagenesis of residues that line this pocket, for example Val77, and in vitro tests of mutated MyoA function would add credit to the structural findings.

The current structure is compared to the structures of isolated PfMyoA and isolated PfAct1 filaments but no comparison with the latest PfMyoA-PfAct1 complex structure, PDB 7ALN, is done. The latter structure is the closest to the one determined by the authors and it is critical that the authors comment on it.

The authors observed very similar fractional buried surface area of MyoA residues in the actin contacting segments with other MyoA-Actin structures, although sequence divergence is high. They also details the similarities and differences for each of aminoacid (Fig. S7) and especially compare their results with the “conserved and ancillary” interface residues defined by Robert-Paganin et al (2021). It is not clear from the text if the latter differences stem from the analysis method or from strudtural divergence. To help the readers understand better, it would be good to include the other plasmodium MyoA structure (PDB: 7aln) in the table of Fig. S7.

Reviewer #3: Here are some suggestions on how the manuscript could be improved, and some comments that I think need clarification.

- Some of the figures are rather hard to follow. For instance, it is really difficult to dissect the interaction between MyoA and Act1 in Fig3, as the assignment of MyoA’s regions is somehow ambiguous. Colour coding like in Fig S6 would make the figure much easier to understand. It is also not at all clear to me what’s the extent of the cavity shown in Fig4, nor what is the role of all the amino acids highlighted in the panels. Moreover, while panels A and B are the same structure, panel B is rotated so it’s hard to relate both snapshots. I find that the cavity appears much clearer in Movie S4. Would a similar representation be possible in the figure? FigS5 does not bring much. Except for the single actin monomer, it is impossible to see whether or not the models fit the density. Movie S2 is much better at that. FigS8 feels unnecessary since it is basically Fig4B but with less information.

- The resolution estimation is somehow inconsistent throughout the manuscript. For instance, the methods state for actomyosin that (l.360-362) “The estimated global resolution reached 3.1 Å calculated with Blocres in Bsoft package version 1.8.6 with a Fourier shell correlation threshold of 0.3)”. In FigS1 however it is clear that this was done with Relion, as for Act1. Could this be just a mix up with the local resolution estimation? The local resolution estimation is confusing as well. For example, the resolution scale in FigS4 goes between 2.1 and 2.5, all values below the average resolution of 2.6 (threshold 0.143). How is this possible? Fig S3 shows the local resolution for actomyosin. In those, the resolution was estimated with a different threshold (0.3). What’s the reasoning behind this?

- A more detailed description of the Act1/MyoA interaction missing. For instance, Fig3 provides the residues from MyoA involved in the interaction, but Act1 is simply shown as surface. This might be important, considering that the residues in these interfaces aren’t strongly evolutionary conserved (as shown in the text), and Act1 is relatively divergent from other actins as well. Similarly, in page 14 the manuscripts states “...forms a second important interface to Act1 [...] that is complementary in shape and hydrophobic in nature. Surprisingly, there are no side chain interactions contributing to this interface, which is rather mediated by main chain atoms”. How can an interaction mediated by H-bonds be hydrophobic?

- X-ray structures of MyoA alone or in complex with its light chains are already available. As with other myosins, there’s small conformational changes upon binding to F-actin. Although these changes are already described in the text (l 204-209), it would be easier to illustrate this with a structural alignment. In addition, there is another MyoA (without light chains) and Act1 structure already available (ref 23). I understand that the preprint for this manuscript precedes Dorit Hanein’s paper (ref 23), but I think a comparison between the two structures would be interesting, and possibly shed some light into the effect of the light chains on myosin. Likewise, in line 170 the manuscript states that “ELC and MTIP bind to the MyoA neck in a compact manner, presumably stiffening the neck region so it acts as a relatively rigid lever arm…”. Isn’t this the case in all myosins? A figure comparing some of the known structures would help strengthening the point.

- The way in which Plasmodium actin breaks ATP and releases Pi seems to be very different than what has been observed in other actin homologues. Having the highest resolution structure of F-actin published to date, it feels as a missed opportunity not to speculate about the structural basis for these differences. Is the configuration of the active site different from other actin filaments? Could the cavity seen in the structure explain the differences in Pi egress kinetics?

- It is not clear to me whether Ser19 is phosphorylated in the structure in this manuscript. Is there independent evidence showing that it is? Was this expected from protein production? The extra density shown in Fig 1 does not look completely convincing. Perhaps doing FDR analysis (with the SPOC package from Carsten Sachse’s group) would help determining whether this is a real feature of the map. If this was indeed a phosphorylation in the map, could the atomic model provide an explanation of how this PTM would affect the function of myosin? On a related note, the manuscript mentions (p. 15) that T417 is not phosphorylated, and contrary to ref 23, a phosphomimetic mutation wasn’t required for proper actin decoration. Any idea why this would be the case?

**Part III – Minor Issues: Editorial and Data Presentation Modifications**

Reviewer #1: (No Response)

Reviewer #2: Lines 211-213: The position of the apicomplexan-specific N-terminal helix links the SH3 and motor domain and contains Ser19. With their structure, the authors are now in a better position to extrapolate how Ser19 phosphorylation status will influence MyoA motor function.

The positioning of Ser15 side chain contradicts the previous findings of its importance for coordinating ADP beta-phosphate. Although this is mentioned in the text, the authors do not comment on the possible explanations of why they observe this difference.

Line 268-269: Sentence starting with :” Within loop 3, Myo 1b has...” needs rephrasing as it is not clear what contacts are described here.

Description in atomic details of the HTH-actin interface would greatly benefit from showing a zoomed-in view of the aminoacid involved.

Reviewer #3: - Line 290 says “Human Cardio-Myopathy” instead of hypertrophic cardiomyopathy.

- In page 16 the text says “samples were prepared on Quantifoil 2/2, 200 mesh air- glowed grids”. The word discharged is missing.

PLOS authors have the option to publish the peer review history of their article (what does this mean?). If published, this will include your full peer review and any attached files.

Reviewer #1: **Yes: **Stephen Matthews

Reviewer #2: No

Reviewer #3: **Yes: **Dr. Felipe Merino

Max Planck Institute for Developmental Biology
---

## [Decision Letter · Decision Letter 1]

18 Feb 2022

Dear Dr. Kursula,

Thank you very much for submitting your manuscript "High-resolution structures of malaria parasite actomyosin and actin filaments" for consideration at PLOS Pathogens. As with all papers reviewed by the journal, your manuscript was reviewed by members of the editorial board and by several independent reviewers. The reviewers appreciated the attention to an important topic. Based on the reviews, we are likely to accept this manuscript for publication, providing that you modify the manuscript according to the review recommendations.

Reviewer # 2 recommends  a revision of the glideosome model and a discussion on the possible rearrangements necessary to fit within the PM-IMC space while reviewer # 3 recommends some trimming

Sincerely,

Dominique Soldati-Favre

Section Editor

PLOS Pathogens

Dominique Soldati-Favre

Section Editor

PLOS Pathogens

Kasturi Haldar

Editor-in-Chief

PLOS Pathogens

orcid.org/0000-0001-5065-158X

Michael Malim

Editor-in-Chief

PLOS Pathogens

orcid.org/0000-0002-7699-2064

Reviewer Comments (if any, and for reference):

Reviewer's Responses to Questions

**Part I - Summary**

Reviewer #1: Paper has been improved and supplemented with new data

Reviewer #2: The revised manuscript from Vahokoski addresses most of the comments and suggestions made by the reviewer. The authors have compared in more details their structure with the published structure from Robert-Paganin et al. (pdbid: 7aln) and added an in vitro experiment measuring the power stroke by optical trapping, showing that the actomyosin light chain complex produces a power stroke similar to other myosins with two light chains. They also modified figures to include additional data and provide more accuracy on certain points.

They have placed GAC within the PM-IMC space, giving an idea of how much space this protein would occupy when connecting actin filaments with the PM. Unfortunately, the illustration presented in figure 2B, with GAC being distant from both the PM and actin, suggests that placement of GAC in incompatible with the model, or that the structure of GAC is not accurate. The text mentions this issue (line 194) but publishing such a model can be confusing for readers "In the parasite, when the additional space taken up by the actin filament (diameter ~6.5 nm) and GAC (Dmax ~25 nm; [16]) is considered, it is clear that there must be significant geometric and steric constraints on how the actomyosin complex might be arranged".

Authors have identified a cavity within the F-actin structure that potentially could be occupied by non-identified solute. In the revised manuscript they discuss disease-associated mutations lining the cavity, supporting the importance of the pocket for F-actin function. Unfortunately, no in vitro or in vivo assays with mutated actins were performs to support their hypotheses.

Phosphorylation status of Ser19 has been addressed by MS, showing a 50% of residues carrying this PTM.

Absence of phosphorylation of Thr417 in the current structure, differing from the 7aln structure where a phosphomimetic Thr417Asp mutation was necessary, is better addressed in the current manuscript

Reviewer #3: The authors have addressed all my previous comments.

The authors essentially addressed all my previous suggestions. All fine from that side. To my eyes, an unfortunate development was that they added an additional experiments, meant to address reviewers 1 and 2, which does not seem to answer any question in the manuscript. They do not exploit those results for much either, so if feels a little pointless. I think the paper would seriously benefit from some trimming.

**Part II – Major Issues: Key Experiments Required for Acceptance**

Reviewer #1: none

Reviewer #2: (No Response)

Reviewer #3: (No Response)

**Part III – Minor Issues: Editorial and Data Presentation Modifications**

Reviewer #1: none

Reviewer #2: (No Response)

Reviewer #3: (No Response)

PLOS authors have the option to publish the peer review history of their article (what does this mean?). If published, this will include your full peer review and any attached files.

Reviewer #1: No

Reviewer #2: No

Reviewer #3: **Yes: **Dr. Felipe Merino

Max Planck Institute for Biology, Tuebingen

Figure Files:

Data Requirements:

Reproducibility:

References:

---

## [Editor Report · Decision Letter 2]

1 Mar 2022

Dear Dr. Kursula,

We are pleased to inform you that your manuscript 'High-resolution structures of malaria parasite actomyosin and actin filaments' has been provisionally accepted for publication in PLOS Pathogens.

Best regards,

Dominique Soldati-Favre

Section Editor

PLOS Pathogens

Dominique Soldati-Favre

Section Editor

PLOS Pathogens

Kasturi Haldar

Editor-in-Chief

PLOS Pathogens

orcid.org/0000-0001-5065-158X

Michael Malim

Editor-in-Chief

PLOS Pathogens

orcid.org/0000-0002-7699-2064
---

## [Editor Report · Acceptance letter]

29 Mar 2022

Dear Dr. Kursula,

We are delighted to inform you that your manuscript, "High-resolution structures of malaria parasite actomyosin and actin filaments," has been formally accepted for publication in PLOS Pathogens.

Best regards,

Kasturi Haldar

Editor-in-Chief

PLOS Pathogens

orcid.org/0000-0001-5065-158X

Michael Malim

Editor-in-Chief

PLOS Pathogens

orcid.org/0000-0002-7699-2064